# Dental Emergencies and Coronavirus Disease-2019: Scoping Review of the Literature and Single Centre Experience

**DOI:** 10.3390/dj10050091

**Published:** 2022-05-20

**Authors:** Agostino Guida, Annamaria Carotenuto, Vladimiro Lanza, Francesco Antonucci, Paola Salerno, Dario Marasca, Umberto Esposito, Maurizio Gargiulo

**Affiliations:** 1U.O.C. Odontostomatologia, A.O.R.N. “A. Cardarelli”, Via A. Cardarelli, 80131 Naples, Italy; lanza.vladimiro@aocardarelli.it (V.L.); francesco.antonucci@aocardarelli.it (F.A.); paola.salerno@aocardarelli.it (P.S.); umberto.esposito@aocardarelli.it (U.E.); 2U.O.C. Chirurgia Maxillofacciale, A.O.R.N. “A. Cardarelli”, Via A. Cardarelli, 80131 Naples, Italy; annamaria.carotenuto@aocardarelli.it (A.C.); maurizio.gargiulo@aocardarelli.it (M.G.); 3Department of Neuroscience, Reproductive Sciences and Dentistry, University of Naples Federico II, Via S. Pansini 5, 80131 Naples, Italy; dario.marasca@gmail.com

**Keywords:** periapical abscess, COVID-19, tooth extraction

## Abstract

Understanding the impact of the COVID-19 pandemic on dental emergencies. A systematic review of the literature (PubMed/Scopus) searching for articles on COVID-19 and dental abscess and a retrospective cohort study with quantitative/qualitative data analysis of our hospital E.R. patients admitted for cervico-facial abscess of dental origin were performed. Thirteen studies could be included in the review, concerning characteristics/management of patients with dental emergencies in hospitals/private practices, generally with poor evidence. For the retrospective analysis, 232 consecutive patients were included (100 study vs. 132 control). The prevalence of dental emergencies (abscess) and relative complications (mediastinitis, exitus) increased. Dental care availability was limited, with strong heterogeneity amongst regions/nations. At-risk (aerosol-generating) procedures were generally avoided, and hospitalization length reduced. Comorbidity patients and males seem less likely to restore regular dentist attendance during the post-lockdown pandemic. Despite the poor scientific evidence, COVID-19 seems to have impacted dental emergencies through limited routine dental care availability and influence on physicians’ and patients’ behaviour.

## 1. Introduction

The coronavirus disease-2019 (COVID-19) pandemic started in December 2019 due to a new member of the *Coronaviridae* family, the Severe Acute Respiratory Syndrome Coronavirus-2 (SARS-CoV-2). Its effects (mainly on the respiratory tract) may be very heterogeneous, varying from mild respiratory symptoms to severe viral pneumonia, leading to respiratory failure and death [1,2].

Since the spread of the disease, unprecedented variations of daily life have been experienced by the global population. Limitations of daily activities (especially during the lockdown period, which in Europe generally happened from March to May 2020) included increased difficulties in receiving dental care.

Since the beginning of the pandemic, the risks of COVID-19 contagion for both operators and patients during dental procedures, especially for high frequencies of aerosol-generating procedures (AGPs), have been highlighted [3]. Droplets and aerosols are commonly produced during dental practice and speech, and coughing and sneezing may occur in a moment during which the patient is not wearing a mask. Long-distance transmission may also occur through viral particles in small droplets carried by the air, causing contagion between patients and dental staff [4,5]. Regional dental practice guidelines tried to limit such occurrences, underlining the importance of personal protective equipment (PPE)—even powered air-purifying respirators (PAPRs) when it is necessary to treat a SARS-CoV-2-positive patient for a dental emergency [6]—and reducing AGPs.

Thus, during the lock down period in Italy, the Ministry of Health advised against routine dental therapies, suggesting the limitation of care to emergencies only (abscess or pain non-responsive to medical therapies), after a telephonic consultation/triage [7]. At the end of May 2020, guidelines for safe dental procedures were published, restoring dental care access; as a matter of fact, such guidelines are still valid. Other countries experienced a similar situation. For example, telephonic triage was recommended in the U.K. in order to provide advice, analgesia, and antibiotic (A.A.A.) prescription if necessary [8]; when insufficient, patients were to be referred to local urgent dental care centers (U.D.C.C.s). The prediction was that the number of dental-related cervicofacial abscess patients presenting to the emergency department would rise due to the impossibility of accessing dental care.

Odontogenic cervicofacial infection/abscess (despite being generally easily avoidable in the absence of severe comorbidities through regular dental care) may be a life-threatening condition, with complications such as airway obstruction, sepsis, the spread of infection, and ultimately death [9].

With COVID-19 being a novel disease, there is limited data regarding the relatively short-term and long-term impact on healthcare in general and consequently on dentistry in particular [3]. During the lockdown period, when the possibility of going to dental practice spontaneously/for prevention was limited, it was inevitable that a certain repercussion would happen in terms of an outbreak of acute dental-related symptoms/conditions. Subsequently, even though the local dental association/ministry of health produced strict guidelines for operating dental care in order to safeguard operators’ and patients’ health against fortuitous contagion from SARS-CoV-2, it was hypothesized that a particular modulation on patients’ attendance to the dental clinic would occur, possibly influencing the outbreak of severe dental complications [3].

Many papers have been published about the impact of COVID-19 on other pathologies. Such papers focus both on the possibility that SARS-CoV-2 infection may influence the outcome of other diseases due to the impact that the ongoing pandemic, with its modifications and limitations on everyday life (e.g., reduction in inpatient availability), may have on other diseases’ management. Results, sometimes reflecting the heterogeneity of COVID-19 management among nations, have been summarized in systematic reviews. [10,11,12,13,14] Similarly, in order to integrate scientific papers focusing on COVID-19 and dental emergencies, we performed a systematic scoping review of the available scientific literature. Furthermore, we performed a retrospective analysis on admissions for dental abscesses during the pandemic at our hospital (“A.O.R.N. A. Cardarelli” hospital—Naples, Campania, Italy), using antecedent years as control. Reviews of scientific literature data about our region (the South of Italy) regarding the impact of COVID-19 on dental emergencies are, according to our knowledge, missing in present literature.

Our work aims to provide an overview of the available scientific evidence of the influence of COVID-19 on dental complications through integration, analysis, and comparison of our databases with the available literature.

## 2. Materials & Methods

### 2.1. Systematic Scoping Review

The “Preferred Reporting Items for Systematic Reviews and Meta-Analyses-PRISMA” checklist was used as a guideline. A systematic scoping review of the present scientific literature was performed during 04/2022 on research engines SCOPUSand PubMed. Research query was: (TITLE-ABS-KEY (“SARS-COV-2”) AND/OR TITLE-ABS-KEY (“COVID-19”)) AND (TITLE-ABS-KEY (“dentistry”) OR TITLE-ABS-KEY (“dental”) OR TITLE-ABS-KEY (“dental abscess”)), designed to be as comprehensive as possible. A first selection was performed through title and abstract analysis (performed by all authors); out of topic papers and/or title/abstract not in English were discarded. Full-text evaluation of the remaining articles was then performed. The excluding criteria were:full text not in Englishduplicate resultsarticle not focusing specifically (in their whole text or in part) on odontogenic abscesses/dental urgencies during the COVID-19 pandemic

Full texts of the eligible articles were then retrieved. Case reports, case series, clinical studies, literature reviews, commentaries, and letters to the editor were included. All results reported were included and summarised textually (performed by all authors).

### 2.2. Single-Center Experience: Evaluation of Dental Abscesses during the COVID-19 Pandemic

The “Strengthening the Reporting of Observational studies in Epidemiology-STROBE” guidelines were followed. Electronic medical records from “A.O.R.N. A. Cardarelli” hospital were retrieved from the database. All consecutive Emergency Room (E.R.) admissions referred to the Dentistry Unit and Maxillofacial Surgery Unit of “A.O.R.N. A. Cardarelli” hospital (Naples, Campania, Italy) from March 2020 to May 2021 were searched for patients presenting odontogenic abscesses. Data regarding date of birth, genre, length of hospitalization, mediastinitis, and exitus were collected. Patients admitted from dental abscess underwent medical therapy (M; ampicillin + sulbactam 1 g + 2 g E.V./die) ± intraoral/percutaneous drainage (D) ± avulsion (A) of the responsible tooth. Avulsion was not performed if the tooth was evaluated as recoverable through endodontic therapy or had already been extracted (abscess persisting after the extraction). For our study, patients were chronologically divided into four groups: lockdown (from March to May 2020—a period of time during which the activities of private dental practices had been blocked in Italy); post-lockdown pandemic (from June 2020 to May 2021); total pandemic (100 patients from March 2020 to May 2021). As a control, the same research was performed for the period antecedent to the pandemic, from January 2018 to February 2020.

Qualitative and quantitative data analysis has been performed to understand if any sensitive changes occurred during the pandemic. Results were evaluated using unpaired Student’s *t*-test (*p*-value) or Pearson Chi-square statistical tests with a 95% confidence interval (CI 95%, *p* < 0.05) to essay the statistical significance of possible variations. All statistical calculations were performed with IBM™ SPSS ver. 21 and Graphpad. As investigated periods had heterogeneous length (3 months for the lockdown periods, 12 months the post-lockdown period, 15 months the whole pandemic, and 26 months the control period), and all data were normalized by day prior to proceed with calculation in order to reduce bias.

## 3. Results

### 3.1. Scoping Review

The screening process is summarised in Figure 1.

The research retrieved 2584 results on Pubmed and 1423 on Scopus. First screening through title/abstract evaluation then produced 27 results. After full text evaluation, 13 papers were finally included:one case report [14]10 clinical studies, nine retrospective [3,15,16,17,18,19,20,21,22] and one prospective [23]one letter to the editor [24]one review paper [6]

All articles retrieved thus presented low scientific evidence, as they were mostly retrospective and/or with a small statistic sample. Furthermore, they were heterogeneous among them, as different regions responded differently to the pandemic (Table 1).

Articles mainly concerned analysis of the quantitative/qualitative characteristics of patients with dental emergencies, the impact of COVID-19 on emergencies in private dental practice, and proposals of protocols for handling dental procedures during the pandemic.

#### 3.1.1. Quantitative/Qualitative Characteristics of Patients with Dental Emergencies

Studies show acute apical periodontitis with or without abscess as the most common cause for patients to go to the dental E.R. during the lockdown [15,16,17,19,21].

Politi and colleagues [15] showed that during the period of closure of dental practices in the United Kingdom (six weeks from March to May), a global reduction of patients presenting to the E.R. was observed. However, a higher percentage (41% vs. 38–39%) of observed patients required hospitalization but with a global reduction of hospital stay—which the authors explained may be due to the physicians’ will to dismiss patients as soon as possible as protection from SARS-CoV-2 infection. Authors conclude that the creation of the Urgent Dental Center (UCC), which provided remote assistance to acute dental patients, managed to reduce the number of patients crowding the hospitals during the pandemic. Similar results were obtained from Long and colleagues [18] in the same geographic area. Kun-Darbois and colleagues [20] also describe similar events in France, where remote support was given to acute dental patients, reducing admission for cellulitis of dental origin by 44% percent. The authors conclude that patients may have avoided hospitals due to the ongoing pandemic.

Studies show different dynamics [16,17,21] in other geographic areas, with a global increase of patients presenting at dental E.R.s with acute dental pain with or without abscess. Eggman and colleagues [21] also highlighted a variation of physicians’ behavior, with reduction of AGPs and increase of teleconsultations. They also highlighted a reduction of patients with comorbidities in the period immediately after the lockdown.

On the other hand, a drastic reduction in global hospital dental care for pediatric patients has been observed [3,22]. Still, dental abscess and acute dental pain were the most common causes of hospital assistance for pediatric patients.

A case report [14] shows an unusual Lemierre Syndrome with brain, lung, and liver involvement of a 24-year-old as a consequence of a dental abscess, which authors relate to the difficulty of receiving dental care during the lockdown.

#### 3.1.2. Impact of COVID-19 on Emergencies Private Dental Practice

A survey administered to private dental practitioners in Northern Italy [23] revealed that about 79.7% of the dentists handled urgencies during the lockdown. Still, a significant reduction in patients was observed, with many dentists (81.2%) providing telephone consultations to evaluate symptoms. Pulpitis and abscesses were the most common urgencies (44.7% and 40.2%, respectively). Protective measures with FP2 masks, surgical gloves, and goggles were used to minimize the risk of contagion both for personnel and patients.

#### 3.1.3. Protocols for Dental Procedures

Yadav and colleagues [24], while a letter to the editor, is the only paper focusing on the impact of COVID-19 on dental emergencies in fragile patients, so it was not discarded. Yadav and colleagues and Levites and colleagues [6] highlight the importance of using PPE in general and the use of PAPR when treating a SARS-CoV-2-positive patient, a possibility when treating an emergency. Telemedicine may help achieve the double objective of taking care of the patients and avoiding hospital crowding and patients undergoing palliative care with life-threatening illnesses (e.g., cancer, chronic heart failure, chronic obstructive pulmonary disease, and cognitive impairment).

### 3.2. Single-Center Experience: Evaluation of Dental Abscesses during the COVID-19 Pandemic

From January 2018 to May 2021, a total of 232 patients (mean age ± SD: 44.2 ± 17.5 years) were admitted through our E.R. due to odontogenic abscesses. Demographic characteristics are summarised in Table 2.

For our study, patients were chronologically divided into four groups: lockdown (28 patients from March to May 2020—a period of time during which the routine activities of private dental practices had been blocked in Italy); post-lockdown pandemic (72 patients from June 2020 to May 2021); total pandemic (100 patients from March 2020 to May 2021); control (132 patients from January 2018 to February 2020). Pearson Chi-square test excluded significant statistical differences for age in every group; as for gender differences, males were admitted significantly more than femalse (*p* = 0.02) in the pandemic group when compared with the control group (no other significant differences were found in the other groups compared to the control group).

Statistical analysis revealed a mean of 13.5 admissions/month (Figure 2).

For dental abscess during the lockdown period (during which private dental practices were generally closed) and 6.1 during the post-lockdown pandemic, with a total pandemic mean of 6.6 admissions/month. Our control period (January 2018–February 2020) showed a mean of 5.3 admissions/month for dental abscesses. Statistical significance (*p* < 0.0001) was found between the lockdown period versus control period and between the lockdown period versus the rest of the pandemic (*p* = 0.0008). No statistical significance was found between post-lockdown/total pandemic when compared with control (*p* > 0.05).

The mean hospital stay (Figure 3) was 11.5 days in the control period, while it was 17.8 during the lockdown and dropped to 8.1 during the post-lockdown pandemic (June 2020–May 2021); all differences among the three periods are statistically significant (*p* < 0.05).

Patients who underwent M, M + A, M + D, and M + A + D (as explained in the Materials and Methods section of the paper) were 35%, 4.5%, 29%, and 44%, respectively, in the control period; 3.5%, 0%, 10%, and 86.5% during the lockdown period; 12.5%, 12.5%, 21%, and 54% in the post-lockdown pandemic period. Differences between these periods are statistically significant according to Chi-square tests (*p* < 0.05). A significant global increase of surgical procedures (A, D) was thus seen during the lockdown and the pandemic period.

Seven mediastinitis and four exitus cases (5% and 3%, respectively) were recorded in the control period, four mediastinitis and one exitus (14–3.5%) during the lockdown period, and four mediastinitis and no exitus (5.5–0%) during the post-lockdown period; the differences between these periods are statistically significant according to Chi-square tests (*p* < 0.05).

To further validate our control cohort, we repeated statistical tests for the number of admitted patients, length of hospital stay, therapies, and complications among the three control years (2018, 2019, Jan–Feb 2020), obtaining no significant statistical differences.

## 4. Discussion

The first result of our scoping review are that, due to pandemic characteristics, the present literature has limited scientific evidence. Differences amongst nations and geographic areas of the same nations exist in dealing with the pandemic and limiting/controlling the spread of cases [25]. For example, COVID-19 spread has been very peculiar in Italy [26]: up until July 2020, the five most-affected regions accounted for more than 75% of cases (Lombardia 39.1%; Piemonte 13%; Emilia Romagna 11.9%; Veneto 8%; Liguria 4.2%; total 76.2%), all of them being in northern Italy. Our region, Campania, which is in southern Italy, accounted for 2% cases (total 4762 cases), unequally distributed among the different cities (Avellino: 561 cases; Benevento: 218; Caserta: 588; Naples: 2693; Salerno: 702). More than half of Campania cases thus occurred in Naples, the most populated area of the region. Despite not being high compared to the rest of our country/other European regions, such numbers resulted in the conversion of 56 Campania hospitals—almost the whole hospital network—into specialized centers for the treatment of COVID-19 (COVID Centers, CCs) [27]. The authors believe that such geographic differences (from state to state and among regions of the same state) in addition to the fact that the pandemic has been ongoing for approximately two years, clearly influenced the quality of the retrieved articles and the data that resulted from our experience. Most of the reviewed articles where retrospective [3,15,16,17,18,19,20,21,22], with one case report [14] and one letter to the editor [24]. The only prospective study was an online survey [23], with 1205 dentists interviewed, which, even with limited evidence due to lack of control, gave an interesting overview of private dental practice during the lockdown. The picture emerging from the survey was that, during the lock down, most dentists (about 80%) did not flinch when facing dental emergency, declaring that they dealt with them with the proper precautions but always prior to telephonic consultations. This confirms the literature data [21] that dental practitioners thoroughly follow directives in hospitals and private practices; AGPs are avoided, and PPE are regularly worn during routine procedures. It is conceivable that this led in many cases to an attempt to deal with the emergency with a first line medical therapy. Yet, in the same study [23], it was reported that the number of urgencies handled weekly during COVID-19 lockdown was lower than that handled in any pre-COVID week. Such data may be interpreted as coherent with the other studies reporting an increase of hospital attendance for dental emergencies.

Retrospective studies—including ours—had some common characteristics. Generally, a quantitative/qualitative analysis of dental emergencies during the lockdown was performed and compared with a control period that was antecedent or posterior to the lockdown period. They showed heterogeneous results, which could have been influenced mainly by how different nations programmed to deal with dental emergencies. In nations where centers for dental emergencies (mostly with telephonic assistance) were organized, a global reduction of hospital accesses was achieved [15,18,20] but with higher percentages of hospitalization required to solve the case. Such results can be interpreted as that only the most complicated/emergency cases (e.g., symptoms/signs—swelling, abscess—not responding to drug therapy) went to a hospital, thus increasing the possibility of necessity of hospitalization in order to complete the therapeutic pathway. In other countries where such an organization was not provided, a global increase of dental E.R. cases occurred, as is also shown by our results. As one study underlines [17], when a hospital provides global dental care and E.R. service, when there is a health situation affecting the global population (even in the pre-covid period), the total number of patients may drop but with an increase in dental emergencies. A general increase in absolute numbers and/or the percentage of dental abscesses was shown, as it was the leading cause of presentation to dental emergency rooms. Such an event may have been caused by limited access to basic dental care/dental practices, which likely caused an increasing rate of complications from dental pathologies (mostly odontogenic abscesses).

When patients with dental abscess report at our E.R., they are dealt with medical therapy and dismissed in the case of resolution with programmed tooth therapy (extraction if necessary). If medical therapy is not effective (likely to an occurred antibiotic resistance), they directly undergo surgery (avulsion ± abscess drainage). Our data show an increase of E.R. access/length of hospitalization for dental abscesses during the lockdown period, as is predictable with routine dental care unavailable, which likely led an increase of serious cases of dental abscess. These high numbers may have been influenced also by the fact that patients, in order to avoid the risk of contagion, may have avoided hospitals unless extremely necessary Similarly, our data show, in addition to an increase of E.R. access for dental abscess, a significant increase of the number of cases dealt with avulsions/surgical abscess drainage during the lockdown and the whole pandemic when compared to the control group. It is conceivable that such an increase of surgical cases may be interpreted as a result of patients presenting to E.R. during the lockdown/pandemic when medical therapy had proven ineffective already, avoiding high-risk public places unless necessary, in order to avoid risk of contagion, even if an official telephonic triage was not provided by healthcare authorities in Italy.

During the lockdown, as other studies from different geographic areas show [16,18,19,20,21,22,23], private practices were generally closed, and this may have caused the augmentation of patients experiencing dental abscess. The increased number in the post-lockdown group may be due to a general fear of going to the dentist, which had similar consequences. Limited access to dental care may thus have led to increased chances of complications due to a dental abscess. As for the case report [10] of Lemierre syndrome in a 24-year-old patient, our study also showed an increased rate of mediastinitis and exitus during the lockdown and the pandemic period in general when compared to control period.

Despite the generally poor evidence of the included studies and our data, another interesting result emerging from our paper is the effect which the pandemic held on both patients’ and physicians’ behavior.

The influence of COVID-19 on the behavior of patients emerges both from our review and our study. Pediatric patients’ [3,22] dental E.R. attendance dropped during the pandemic, possibly because dental abscess in pediatric patients rarely requires hospital procedures. Still, in proportion to reduced number of visits, the percentage of dental abscess/emergencies seemed to rise in pediatric patients too [22]. Another study [21] shows general reticence for comorbidity patients to go to dental hospitals during the pandemic unless they are experiencing severe conditions, and our study showed a prevalence of male attendance in the same period, significantly higher than the control period. As dental care is fully accessible in our geographic area, this may be interpreted as a lack of reprise of dentist attendance by male patients. Such limited data suggests that some categories feel less confident/safe than others in attending dental practices regularly during the post-lockdown pandemic. As a matter of fact, behavioral changes between male and female and among different ages has been observed as a consequence of the pandemic [27]. Similarly, our results and other articles show an influence on physicians’ behavior [16,18,20,21,22]. A general tendency in anticipating the discharge of the patient’s stay may be observed, possibly to avoid unnecessary risk of SARS-CoV-2 contagion during hospitalization. Consequently, the aftermath on dental team mental health has been underlined by the literature [28]. As the pandemic has been compared to World War II in terms of threat to the mental health of healthcare personnel, stress management is advised for dental team members at any time, inside and outside the workplace.

Conclusively, it is important to underline how results provided in our scoping review arise from data (including ours) with limited scientific evidence and should thus be considered with due precautions. Prospective multicenter studies would be needed to fully comprehend COVID-19′s influence on dental issues. COVID-19 seems to influence dental emergencies differently in different geographic regions, reflecting the status of the organization of public dental care. The organization of telephonic dental emergency centers seems an effective way to prevent E.R. overload when primary center/private practice are unavailable. Such experience may in the future be used in order to lighten dental E.R. burden. Yet, when routine dental care is not fully accessible, a general increase of dental abscess incidence may be observed, with consequent increase of exitus and mediastinitis rates. COVID-19 also seems to influence both physicians’ behavior (reduction of AGPs, reduction of length of hospitalization) and patients’ behavior. Clinicians, despite frequent changes in guidelines as scientific evidence evolves [29], seem reactive and capable of adapting, both in primary (private practices) and secondary (hospitals) care centers, providing healthcare and protecting themselves and patients from SARS-CoV-2 contagion. Some categories (comorbidity patients, males) seem indeed less inclined to restore regular dental care, thus paradoxically exposing themselves to a higher risk of complications and hospitalization, reproducing the same situation which occurred during the lockdown phase of the pandemic, when the absence or limited availability of primary care dentistry led to an increased demand for emergency dental treatments [30]. Moreover, it occurs more and more often that increasing sections of secondary/tertiary care centers/hospitals have to be reserved to patients incidentally found SARS-CoV-2 infected at pre-hospitalization swab, even if they were referred for non-Covid-related reasons and were completely asymptomatic. It is thus desirable that, in the near future, projects of home health care and monitoring will be implemented, in order not to reduce non-Covid inpatients.

## Figures and Tables

**Figure 1 dentistry-10-00091-f001:**
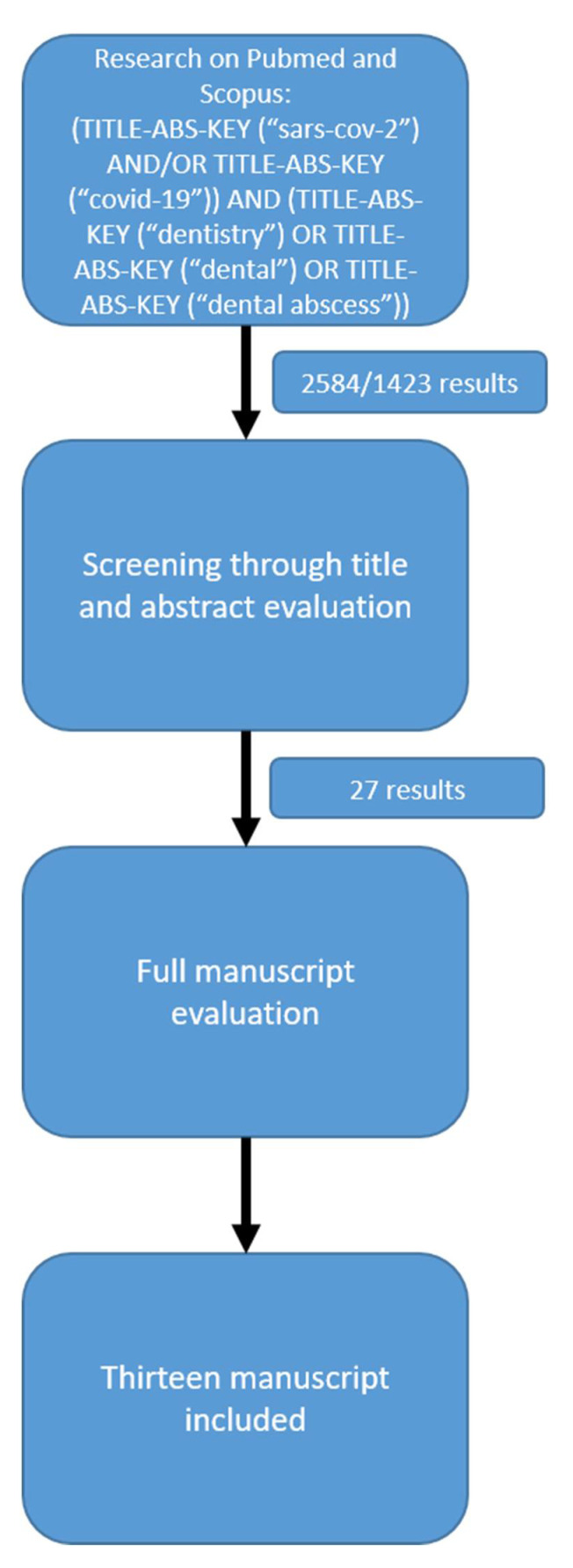
Articles screening process.

**Figure 2 dentistry-10-00091-f002:**
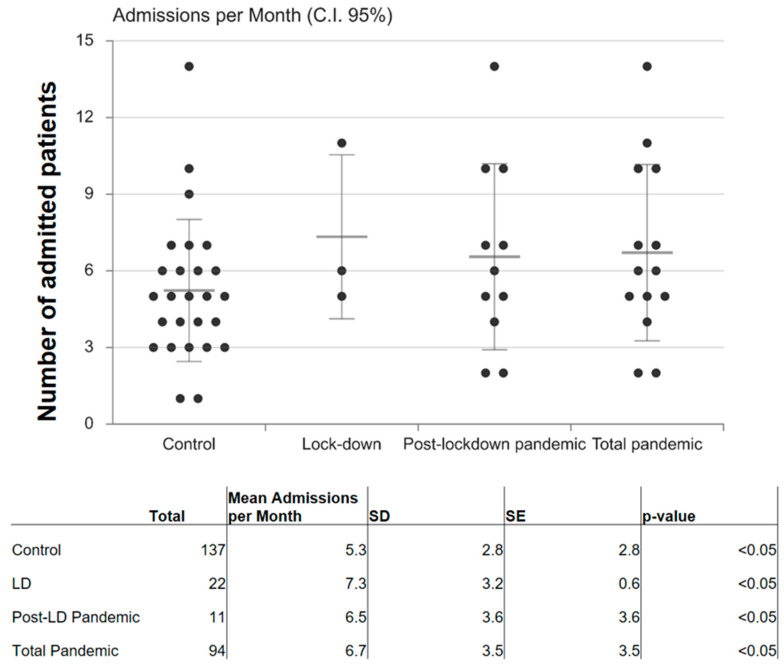
Dot plot of numbers of admissions per month; 95% confidence interval; *p* < 0.05. CI: confidence interval; SD: standard deviation; SE: standard error.

**Figure 3 dentistry-10-00091-f003:**
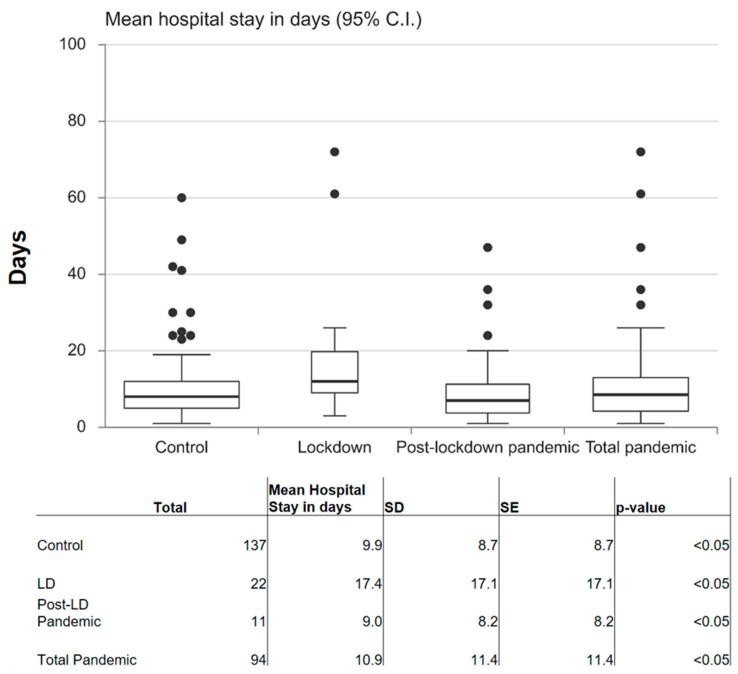
Box plot of hospital stay in days; 95% confidence interval; *p* < 0.05. CI: confidence interval; SD: standard deviation; SE: standard error.

**Table 1 dentistry-10-00091-t001:** Included articles (in alphabetical order).

Article	Article Type	Main Focus
Alzahrani et al. [22]	Retrospective clinical study	Characteristics of patients with dental emergencies
Eggman et al. [21]	Retrospective clinical study	Characteristics of patients with dental emergencies
Howley et al. [14]	Case report	Characteristics of patients with dental emergencies
Kun-Darbois et al. [20]	Retrospective clinical study	Characteristics of patients with dental emergencies
Levites et al. [6]	Review	Protocols
Long et al. [18]	Retrospective clinical study	Characteristics of patients with dental emergencies
Petrescu et al. [16]	Retrospective clinical study	Characteristics of patients with dental emergencies
Politi et al. [15]	Retrospective clinical study	Characteristics of patients with dental emergencies
Ramirez et al. [19]	Retrospective clinical study	Characteristics of patients with dental emergencies
Salgarello et al. [23]	Prospective study	Impact of COVID-19 on private dental practice
Ustun et al. [3]	Retrospective clinical study	Characteristics of patients with dental emergencies
Yadav et al. [24]	Letter to the editor	Protocols
Yu et al. [17]	Retrospective clinical study	Characteristics of patients with dental emergencies

**Table 2 dentistry-10-00091-t002:** Demographic characteristics of the study population. SD: standard deviation; M: medical therapy—see text; A: tooth extraction; D: intraoral/percutaneous drainage; E: exitus; MED: mediastinitis.

	Number of Patients	Age (Mean ± SD)	Sex (Female/Male)	Therapy (M/M + A/M + D/M + A + D)	Complications (MED/E)
Lockdown (March–May 2020)	28	46.9 ± 15.8	14/14	1/5/2/20	4/1
Post lockdown pandemic (May 2020–May 2021)	72	44.2 ± 16.2	28/44	9/9/15/39	4/0
Total pandemic (March 2020–May 2021)	100	45.5 ± 16.1	36/64	10/14/17/59	8/1
Control (January 2018–February 2020)	132	43.8 ± 17.6	67/65	46/8/37/41	7/4
Total	232	44.2 ± 17.5	145/87	56/22/54/100	15/5

## Data Availability

Data may be disclosed on request to the corresponding author.

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
