# Peer review of "Dental Emergencies and Coronavirus Disease-2019: Scoping Review of the Literature and Single Centre Experience"

_dentistry, 2022, doi:10.3390/dj10050091_

Round 1

Reviewer 1 Report

Dear authros,

the present manuscirpt aims to describe the possible emergencies experiences related to Covid-19 and dental treatments.The manuscript is original and well conducted.

from my point of view is acceptable for publication on this journal.

Author Response

Dear Reviewer,

We are happy you found our paper suitable for pubblication. We are grateful for your positive response.  On behalf of all the authors,

Kind regards,

Agostino Guida, DMD, PhD, MSc

Reviewer 2 Report

The manuscript "Dental emergencies and Coronavirus disease-2019: scoping review of the literature and single centre experience." is a well-written manuscript that provides data regarding the management of dental emergencies during COVID-19 pandemic.

In my opinion, some changes must be made throughout the manuscript before being published.

1. There are some terms and phrases that are not properly used in english and should be improved, like:
"dental procedures have been identified as vulnerable for 38 COVID-19 contagion"
"Viral particles and small infected droplets in 41 the air may cause through long-distance transmission between the patient and the dental staff "
"such guidelines are still in force."
"showing at E.R. " (r. 255)
"
the behavioral effect of the pandemic." (r. 267)
"implanted in order not to reduce non-Covid inpatients." (r. 296)
"latter to editor". (r. 136) and so on.

2. In the Introduction and in the Materials and Methods sections, from my point of view, there must be made clear the region/country/county the study was made in.

Author Response

Dear Reviewer,

we are happy you have found our manuscript worthy. Thank you for your precious suggestions. We have followed your corrections thoroughly; please tell us in case additional changes should be made. Here is a point by point response to your letter:

1) English has been improved for the requested sentences:

  • "dental procedures have been identified as vulnerable for 38 COVID-19 contagion" has been modified in "it has been highlighted how there is a risk of COVID-19 contagion both for operators and patients during dental procedures;
  • "Viral particles and small infected droplets in the air may cause through long-distance transmission between the patient and the dental staff" has been modified in "Long-distance transmission may also occur, through viral particles in small droplets carried by the air, causing contagion between patients and dental staff ";
  • "such guidelines are still in force" has been modified in "such guidelines are still valid";
  • "showing at E.R." has been modified in "presenting to E.R.";
  • "the behavioral effect of the pandemic." has been modified in "the effect which the pandemic held on both patients’ and physicians’ behavior";
  • "implanted in order not to reduce non-Covid inpatients." has been modified in "projects of home health care and monitoring will be implemented, in order not to reduce non-Covid inpatients";
  • "latter to editor" has been modified in "letter to the editor".

2) We specified both at the end of the Introduction and in M&M that the study was performed at “A.O.R.N. A. Cardarelli” hospital - Naples, Campania, Italy.

On behalf of all the authors,

Kind regards

Reviewer 3 Report

Thank you for submitting the paper Dental emergencies and Coronavirus disease-2019: scoping review of the literature and single centre experience.

 Thank you for the opportunity to review the above referenced manuscript.

Quality of figure 1 is not good enough.

 Even though the manuscript was well written but the reviewer could not find the novelty of this work. Lot of papers about covid disease and dental emergencies have been published until now, which draws less attention due to the lack of originality.

 Thank you again for the opportunity to assist Dentistry Journal.

Author Response

Dear Reviewer,

thanks for your corrections. We followed your suggestions thoroughly; in case it is not sufficient, please do not hesitate to ask for further modifications. Here is a point by point response to your letter:

1)We improved quality of figure 1;

2)We apologise if the purpose of our study was unclear; we added this paragraph at the end of the "Introduction" section: 

"Many papers have been published about the impact of COVID-19 on other pathologies. Such papers focus both on the possibility that SARS-COV-2 positivity may influence the outcome of other diseases both on the impact that the ongoing pandemic, with its changings and limitations on everyday life (e.g.: reduction in inpatients availability) hold other diseases management. Results, sometimes reflecting heterogeneity of COVID-19 management among nations, have been summarized in systematic reviews. Similarly, in order to integrate scientific papers focusing on COVID-19 and dental emergencies, we performed a systematic scoping review of available scientific literature. Furthermore, we performed a retrospective analysis on admission for dental abscess during the pandemic at our hospital (“A.O.R.N. A. Cardarelli” hospital - Naples, Campania, Italy), using antecedent years as control. Review of scientific literature/data about our region (the South of Italy) regarding the impact of COVID-19 on dental emergencies are, limited to our knowledge, missing in present literature.  Our work aims to provide an overview of the scientific evidence of the influence of COVID-19 on dental complications through analysis and comparison of our databases with the available literature."

Here are the scientific references (that we added to our paper):

  1. Ghazanfar, H. et al. (2022). Impact of COVID-19 on the Gastrointestinal Tract: A Clinical Review. Cureus, 14(3)
  2. Sabaghian, T. et al. (2022). COVID-19 and Acute Kidney Injury: A Systematic Review. Frontiers in medicine, 9, 705908
  3. Vinayagamoorthy, K. et al. (2022). Prevalence, Risk Factors, Treatment and Outcome of multidrug resistance Candida auris Infections in Coronavirus Disease (COVID-19) Patients: A Systematic Review. Mycoses, 10.1111/myc.13447. Advance online publication
  4. Farsi, Y. et al. (2022). Diagnostic, Prognostic, and Therapeutic Roles of Gut Microbiota in COVID-19: A Comprehensive Systematic Review. Frontiers in cellular and infection microbiology, 12, 804644.

We hope that you will now enjoy our paper which is, as far as we know, the first review on COVID-19 and dental emergencies and the first report on dental emergencies during the pandemic of our region (the South of Italy). On behalf of all the authors,

Kind regards

Agostino Guida, DMD, PhD, MSc

Reviewer 4 Report

Title: Appropriate 

Abstract:   PubMed/Scopus  or PubMed and Scopus 

Kindly check with this

Keywords: Mesh Terms

Introduction:

What is the need for this study?

Methods:

Kindly submit the search documents as a supplementary file in Scopus and Pub Med

Actually, the lockdown started in March -2020, So I would recommend the authors extend the search till December 2021 so that you could have a potential articles 

Results:

letter to the editor may not be your choice for scoping review 

table 2 a lot of confusion please explain in point of point to avoid confusion 

Discussion:

Use the below couple of manuscripts for discussion 

Woolley, J., & Djemal, S. (2021). Traumatic Dental Injuries During the COVID-19 Pandemic. Primary dental journal10(1), 28–32. https://doi.org/10.1177/2050168420980994

Yang, Y. T., Zhang, W., Xie, L., Li, Z. B., & Li, Z. (2020). Characteristic changes of traumatic dental injuries in a teaching hospital of Wuhan under transmission control measures during the COVID-19 epidemic. Dental traumatology : official publication of International Association for Dental Traumatology36(6), 584–589. https://doi.org/10.1111/edt.12589

Tonkaboni, A., Amirzade-Iranaq, M. H., Ziaei, H., & Ather, A. (2021). Impact of COVID-19 on Dentistry. Advances in experimental medicine and biology1318, 623–636. https://doi.org/10.1007/978-3-030-63761-3_34

Shah, A., Bryant, C., Patel, J., Tagar, H., Akintola, D., & Obisesan, O. (2020). COVID-19: establishing an oral surgery-led urgent dental care hub. British dental journal228(12), 957–963. https://doi.org/10.1038/s41415-020-1713-5

Ellwood F. (2021). Dental Emergencies: Perceived impact of the COVID-19 pandemic on the mental health and wellbeing of dental teams in the UK. Primary dental journal10(3), 63–68. https://doi.org/10.1177/20501684211029425

Bhumireddy, J., Mallineni, S. K., & Nuvvula, S. (2021). Challenges and possible solutions in dental practice during and post COVID-19. Environmental science and pollution research international28(2), 1275–1277. https://doi.org/10.1007/s11356-020-10983-x

Alzahrani, S. B., Alrusayes, A. A., Alfraih, Y. K., & Aldossary, M. S. (2021). Characteristics of paediatric dental emergencies during the COVID-19 pandemic in Riyadh City, Saudi Arabia. European journal of paediatric dentistry22(2), 95–97. https://doi.org/10.23804/ejpd.2021.22.02.2

The discussion needs to be improved.

Limitations of your scoping review should be addressed at the end of your discussion.

Conclusion: Objectiev based.

Reference: Last Reference was not stated properly 

Author Response

Dear Reviewer,

Thank you for your precious corrections. We performed the changings you requested; please do not hesitate to ask for further modifications in case what we performed is insufficient. Here is a point by point response to your letter:

1) We corrected in "PubMed/Scopus";

2) We replaced the keywords with Mesh Terms;

3) We apologise if the purpose of our study was unclear; we added this paragraph at the end of the "Introduction" section:

"Many papers have been published about the impact of COVID-19 on other pathologies. Such papers focus both on the possibility that SARS-COV-2 positivity may influence the outcome of other diseases both on the impact that the ongoing pandemic, with its changings and limitations on everyday life (e.g.: reduction in inpatients availability) hold other diseases management. Results, sometimes reflecting heterogeneity of COVID-19 management among nations, have been summarized in systematic reviews. Similarly, in order to integrate scientific papers focusing on COVID-19 and dental emergencies, we performed a systematic scoping review of available scientific literature. Furthermore, we performed a retrospective analysis on admission for dental abscess during the pandemic at our hospital (“A.O.R.N. A. Cardarelli” hospital - Naples, Campania, Italy), using antecedent years as control. Review of scientific literature/data about our region (the South of Italy) regarding the impact of COVID-19 on dental emergencies are, limited to our knowledge, missing in present literature. Our work aims to provide an overview of the scientific evidence of the influence of COVID-19 on dental complications through analysis and comparison of our databases with the available literature."

Here are the scientific references (that we added to our paper):

  1. Ghazanfar, H. et al. (2022). Impact of COVID-19 on the Gastrointestinal Tract: A Clinical Review. Cureus, 14(3)
  2. Sabaghian, T. et al. (2022). COVID-19 and Acute Kidney Injury: A Systematic Review. Frontiers in medicine, 9, 705908
  3. Vinayagamoorthy, K. et al. (2022). Prevalence, Risk Factors, Treatment and Outcome of multidrug resistance Candida auris Infections in Coronavirus Disease (COVID-19) Patients: A Systematic Review. Mycoses, 10.1111/myc.13447. Advance online publication
  4. Farsi, Y. et al. (2022). Diagnostic, Prognostic, and Therapeutic Roles of Gut Microbiota in COVID-19: A Comprehensive Systematic Review. Frontiers in cellular and infection microbiology, 12, 804644.

4) We uploaded for you pdf files of the reviewed articles (as a single pdf, as the system unfortunately does not allow multiple file submission from the authors to the reviewer);

5) We re-performed the article websearch in order to update results as you suggested; the work from Alzahrani et al. was added;

6) As you suggested, we underlined that this is only a letter to the editor, but we preferred to keep it as it is the only article focusing on Fragile Patients; we underlined this in the results section;

7) We improved the in-text explaination of table 2;

8) We improved "Discussion" adding the references you suggested, underlining the limitation of the scoping review; the work from Alzahrani et al. was also added to the review;

9) We corrected the reference.

We heartfully thank you for the corrections which we believe arose the level of our article. On behalf of all the authors,

Kind regards

Reviewer 5 Report

This is a review on dental emergencies and hospitalization during covid-19 lockdown. The topic is relevant and the data is interesting. Understanding the state of dental care during lockdown and post lockdown is important to estimate the consequences of such change on the dental healthcare and to increase preparedness going ahead. There are minor and few major revisions that need to be done.

Minor

  1. Full form of APG, PARP, AGP, STROBE
  2. Rewrite paragraph starting from line 114 to line 120. There are errors related to number of publications included and spelling errors
  3. Include the criteria used to shortlist 12 studies out of 27 studies.
  4. English language and spelling need to be improved.
  5. Drainage spelling in table 2 legend is wrong

Major

For Figure 2 show all the individual observations to understand the distribution. Instead of bar plot make a dot plot with the standard deviation/error and p value on the graph.

For Figure 2 the control group and the post lockdown are over 1 year interval while the lockdown group is only less than 3 months. It will be important to normalize this data by time/days to make it more unbiased.

Logically the effect of COVID19 emergency on dental healthcare should have been seen in post lockdown period since only emergency dental care was offered in most of the locations, Authors need to explain why they see increase in dental emergency hospitalizations (admissions and length of stay) during COVID lockdown.

Author Response

Dear Reviewer,

We are happy you were satisfied with our work. We followed your corrections thoroughly; please do not hesitate to ask for further modifications in case you are not fully satisfied with our work. Here is a point by point response to your letter:

Minor

1) We apologise but "APG" was a typing error for "AGP"; we inserted the full form for every acronym at its first use;

2) We have corrected the passage;

3) The passage explaining selection criteria in "Materials and Methods" section was indeed unclear; we clarified such passage;

4) We performed a general English revision;

5) We corrected the spelling;

Major

1) Graph have been remade and split in two in order to make it nore intellegible; please note that for the second one a box-plot was preferred because otherwise, due to the big numbers of data, it would have been unclear.

2) All calculations were already made by normalizing data by time; we aplogise if it was unclear and clarified it furtherly in Materials & Methods;

3) We apologise if this passage was unclear and we clarified it furtherly in the discussion section.

On behalf of all the authors, 

Kind regards

Round 2

Reviewer 3 Report

The paper has improved. Now it is suitable for publication.

Thank you so much. 

Author Response

Dear Reviewer,

We are happy you are satisfied with our work. On behalf of all the authors,

Kind regards

Reviewer 4 Report

All the queries have been addressed.

The submitted manuscript has not used the template of the journal.

Change the format and highlight the modifications.

The manuscript looks acceptable 

Author Response

Dear Reviewer,

We apologise about formatting, but we have double checked and on our PCs the changes appear highlighted. We have re-uploaded the word file, but if you still see changes without highlighiting maybe we are encountering some issues with system compatibility. Anyway, we are happy you are satisfied with our work. On behalf of all the authors,

Kind regards

Reviewer 5 Report

I would like to accept the revised manuscript.

Author Response

(The authors gave the same response as above.)
